# Antibacterial Effect of Low-Concentration ZnO Nanoparticles on Sulfate-Reducing Bacteria under Visible Light

**DOI:** 10.3390/nano13142033

**Published:** 2023-07-09

**Authors:** Hua Yang, Jialin Zhang, Zhuoran Li, Jinrong Huang, Jun Wu, Yixuan Zhang, Honghua Ge, Yuzeng Zhao

**Affiliations:** Shanghai Key Laboratory of Materials Protection and Advanced Materials in Electric Power, Shanghai Engineering Research Center of Energy-Saving in Heat Exchange Systems, Shanghai University of Electric Power, Shanghai 200090, China; 18616043993@163.com (H.Y.); zjl80272021@163.com (J.Z.); 15981756602@163.com (Z.L.); hjinrong2021@163.com (J.H.); wujun4368@163.com (J.W.); zyx_zhangyixuan@163.com (Y.Z.); zhaoyuzeng@shiep.edu.cn (Y.Z.)

**Keywords:** ZnO nanoparticles, sulfate-reducing bacteria, antibacterial activity, biofilm

## Abstract

The effect of ZnO nanoparticles (ZnO NPs), with different concentrations in simulated water, on the activity of sulfate-reducing bacteria (SRB) and their adhesion behaviour on stainless-steel surfaces, with and without visible light treatment, were investigated. The results showed that the concentration of ZnO NPs and light treatment greatly influenced the antibacterial performance of the NPs. In the water solution without light treatment, the low concentration (no more than 1 mg/L) of ZnO NPs in the aqueous solution promoted the growth of SRB, and the amount of biofilm attached to the stainless-steel surface increased. As the concentration increased, ZnO NPs exhibited antibacterial effects. In water under visible light irradiation, ZnO NPs showed antibacterial performance at all the concentrations studied (0.5~50 mg/L), and the antibacterial efficiency increased with the increase in the concentration of NPs. The determination results of the reactive oxygen species showed that light treatment can stimulate ZnO NPs in water to generate ·OH and O_2_^·−^, which exhibited good antibacterial properties. The adhesion amount of SRB on the stainless-steel surface was inversely proportional to the antibacterial efficiency of ZnO NPs.

## 1. Introduction

There are various microorganisms widely present in industrial cooling water, which pose great harm to industrial production. For example, sulfate-reducing bacteria (SRB) in water, whose metabolite is H_2_S, can accelerate metal corrosion and increase the occurrence of accidents. The adhesion of biofilms on the surface of heat exchange tubes can reduce heat transfer efficiency and result in significant economic losses; therefore, it is necessary to control microorganisms in water. Traditional methods use fungicides for microbial control [1], which not only causes environmental pollution but also leads to microbial resistance. In contrast, physical sterilisation methods such as ultraviolet radiation and electromagnetic fields do not pollute the water environment, but there are problems such as low sterilisation rates and the incomplete removal of biofilms. In recent years, metal oxide nanoparticles have received significant attention due to their unique antibacterial properties. These oxide nanoparticles have greater durability, lower toxicity, and higher stability [2]. Metal-based NPs (metal and metal oxide nanoparticles) have a variety of types and are more promising nanomaterials with significant research.

ZnO nanoparticles (NPs) are common metal oxide nanomaterials with excellent antibacterial and photocatalytic properties, and they have good application prospects in drug treatment, food preservative packaging, etc. Due to the special surface properties and physicochemical properties, ZnO NPs can induce the reduction in bacterial activity and destroy the integrity of cell membranes, thereby achieving an antibacterial effect. The antibacterial properties of nanoparticles are related to their properties (such as concentration [3], particle size [4], surface charge [5], etc.), microbial species [6], and environmental conditions [7]. Studies so far have shown that ZnO NPs have toxic effects on various bacteria such as *Escherichia coli* [8], *Bacillus subtilis* [9], *Pseudomonas aeruginosa* [10], etc. In the past years, the research on the antibacterial effect of ZnO NPs was mostly focused on pathogenic bacteria, while little research was conducted on the effect of corrosive, sulfate-reducing bacteria (SRB). Recently, Li et al. [11] studied the effect of a high concentration of ZnO particles on the adhesion of SRB to carbon steel; they confirmed that 300 mg/L of ZnO particles could inhibit the adhesion of SRB on the surface of carbon steel and reduce the formation of biofilm, thereby inhibiting the corrosion of carbon steel.

In previous studies on the antibacterial effect, the concentration of ZnO NPs used was mostly within 100 mg/L. Under this concentration range, ZnO NPs inhibited most strains. Lower concentrations such as 5 mg/L and 15 mg/L can inhibit the growth of *Klebsiella pneumoniae* [12] and *Escherichia coli* [13], respectively. In addition, the antibacterial effect of ZnO NPs is enhanced under visible light. Zudyte et al. [14] found that ZnO NPs reduced the number of *E. coli* by about two orders of magnitude under the excitation of visible light compared to that of the control group. Illumination can reduce the concentration of nanoparticles used.

In this paper, the antibacterial properties and mechanisms of low-concentration ZnO NPs of SRB under visible light in simulated water were studied, and the effect of low-concentration ZnO NPs on the adhesion of SRB on stainless steel surface was discussed.

## 2. Experimental

### 2.1. Experimental Materials and Media

ZnO NPs were purchased from Aladdin Co., Ltd. (Shanghai, China). ZnO NPs were added in simulated water (the composition is shown in Table 1) and sonicated for 30 min to prepare the ZnO nanofluid. Grade 304 stainless steel (the composition is listed in Table 2) was used for biofilm attachment experiments.

Using simulated water as the base medium, its composition (Table 1) referred to the supplementary water for the circulating cooling water of a power plant in China.

The experimental solutions were ZnO nanofluids composed of ZnO nanoparticles and simulated water that were inoculated with SRB medium at a ratio of 1:10 (*V*/*V*). The solution was sealed with sterilized liquid paraffin to maintain the anaerobic environment. The experimental visible light source was a PLS-SXW300 xenon lamp (Beijing Perfectlight Technology Co. Ltd., Beijing, China) with a visible light output power of 19.6 W and a light intensity of 1000 mA/cm^2^, with a 420 nm filter. The light source was 25 cm away from the sample during the irradiation process, and the light treatment time was 2 h. The treated water samples were then used for the subsequent experiments.

### 2.2. Bacterial Strains and Culture Condition

The SRB strain used in this paper was obtained from the sludge of a pond, then extracted and purified. The sample microbial population analysis showed that the strain belonged to *Desulfovibrio*.

The culture medium of SRB consisted of 0.2 g/L of MgSO_4_·7H_2_O, 0.01 g/L of KH_2_PO_4_, 10 g/L of NaCl, 0.2 g/L of (NH_4_)_2_Fe(SO_4_)_2_, 4 mL/L of Sodium lactate, 1 g/L of yeast extract, 0.1 g/L of vitamin C, and it was maintained at pH 7.0~7.2. The culture medium was sterilized in a high-pressure sterilisation pot, cooled, and then transferred to the bacterial strain for cultivation at 35 °C. the SRB bacterial solution was inoculated into different concentrations of ZnO nanofluids, and experiments were conducted under light and no light conditions. After 2 h, a certain amount of water sample was taken, and the dilution plate counting method was used to determine the bacterial content. After 48 h, the bacterial colonies were counted, and the antibacterial efficiency was calculated according to Equation (1).
(1)$$\mathrm{Inhibition}\%=\frac{\mathrm{colonies}\;\mathrm{in}\;\mathrm{control}\;\mathrm{samples}-\mathrm{colonies}\;\mathrm{in}\;\mathrm{experimental}\;\mathrm{samples}}{\mathrm{colonies}\;\mathrm{in}\;\mathrm{control}\;\mathrm{samples}}\times 100\%$$

### 2.3. Determination of Zn^2+^ in Water

The concentration of Zn^2+^ in the supernatant of ZnO nanofluid was determined by atomic absorption spectrophotometry [15], using an atomic absorption spectrophotometer (TAS-990, Beijing Puxi General Instrument Co. Ltd., Beijing, China).

### 2.4. Measurement of Reactive Oxygen Species (ROS)

The three-dimensional excitation–emission matrix (EEM) fluorescence spectrum corresponding to ·OH was obtained using a fluorescence spectrophotometer (RF-5301pc, Shimadzu, Kyoto, Japan) with terephthalic acid as the fluorescence capture agent. The prepared NPs’ suspension was measured colourimetrically at the wavelength of 530 nm. The O_2_^·−^ concentration was calculated according to the standard curve of NaNO_2_ concentration–absorbance [16].

### 2.5. Analysis of the Cell Membrane Integrity

When the cell membrane ruptures, intracellular DNA, RNA, and other macromolecules flow out from the membrane into the solution. These substances have an absorption peak of 260 nm [17]. The samples were centrifuged (10,000× *g*) for 5 min at 4 °C and the integrity of the cell membrane was measured by the absorbance of the supernatant with a UV-2700 UV-Vis spectrophotometer (Unico (Shanghai) instrument Co. Ltd., Shanghai, China) [18]. The untreated, bacteria-containing simulated water was used as a reference in the determination.

### 2.6. Surface Analysis

ZnO NPs used in this paper were characterized using transmission electron microscopy (TEM, JEM 2100F, FEOL Ltd., Tokyo, Japan). The SRB-attached stainless-steel samples were first fixed with 2% glutaraldehyde, then dehydrated with 50, 70, 80, 90, 95, and 100% ethanol successively. After drying, a scanning electron microscope (SEM) (SU-1500, Hitachi Ltd., Tokyo, Japan) with an EMAX energy dispersive spectrometer (EDS) (Hitachi Ltd., Tokyo, Japan) was used to observe and analyse the adhesion of biofilms on the surface.

## 3. Results and Discussion

### 3.1. Antibacterial Effect of ZnO NPs on SRB

Figure 1 is a TEM image of ZnO NPs. It can be observed that the particle size of most ZnO NPs is about 25 nm. ZnO NPs are considered as potential antimicrobial agents for microorganisms. In order to explore its toxicity to SRB, antibacterial activity under different conditions was investigated using the dilution plate counting method. The antibacterial efficiency of ZnO NPs’ concentration in solutions with or without light treatment is shown in Figure 2. Without light treatment, when the concentration of ZnO NPs was lower than 1 mg/L, the existence of ZnO NPs promoted the growth of SRB. When the concentration of ZnO NPs reached 1 mg/L or above, the quantity of SRB in water decreased, and the antibacterial efficiency increased. When the concentration of ZnO NPs reached 10 mg/L and 50 mg/L, the antibacterial efficiency was 28.30% and 55.03%, respectively. In simulated water without ZnO NPs, the light treatment had an inhibitory effect on the growth of SRB. The antibacterial efficiency was 9.32%, which may be because the wavelength at 420 nm is close to violet light, which can induce genetic mutation and kill SRB [19]. The antibacterial ability of ZnO NPs was greatly improved under illumination, and the antibacterial efficiency reached 61.01% and 92.11% at the ZnO NPs’ concentration of 10 mg/L and 50 mg/L, respectively. This is because ZnO NPs have a small particle size and a large specific surface area. So, the higher the concentration, the larger the contact area is between ZnO NPs and microorganisms, the more complete the reaction, and the more significant the bactericidal effect. Under lighting conditions, the higher the concentration is for ZnO NPs, the more charge carriers and ROS will be generated.

The antibacterial effect of ZnO NPs can be explained by three mechanisms: (1) Contact with cells—when nanoparticles contact bacteria, the bacterial physical form can be damaged. When ZnO NPs contacted *Escherichia coli*, the cell wall ruptured, and the cell membrane decomposed [20]. (2) The dissolution of Zn^2+^—the Zn^2+^ released from ZnO NPs in water can be adsorbed on the surface of negatively charged bacteria, inhibiting the synthesis and metabolism of proteins, and interfering with the growth of microorganisms. Zn^2+^ with a concentration higher than 50 mg/L would affect the activity of enzymes in nitrifying bacteria and inhibit the nitrification process [21]. (3) The production of ROS—ZnO NPs have optical properties, and the ROS generated by illumination can cause a series of irreversible damages such as bacterial oxidative stress and loss of enzymatic activity. Appierot et al. [22] thought that the antibacterial effect of ZnO NPs was mainly caused by the damage of ROS to the cell membrane. The bandgap width of ZnO NPs is usually between 3.2 and 3.5 eV [23], and we calculated that the bandgap width of the ZnO NPs used in this paper was 3.22 eV. Due to its inherent defects, the ZnO NPs can be activated not only by ultraviolet light but also by visible light. Although the absorption rate is low, the excitation below the bandgap produces an appropriate amount of long-lived excitons, resulting in additional ROS [14]. Wang et al. [24] demonstrated that the presence of oxygen vacancies in ZnO NPs can effectively expand the visible light absorption range of ZnO and improve its visible light photocatalytic efficiency. Lipovsky et al. [25] found that when ZnO NPs’ suspension was irradiated with visible light in the range of 400–500 nm, the level of oxygen radicals significantly increased and singlet oxygen was produced. Prasanna’s research [26] shows that under visible light irradiation, oxygen vacancies on the surface of semiconductor ZnO NPs can capture photoexcited electrons, increasing the production of ROS.

According to the above antibacterial mechanisms of nanoparticles, the actions of ZnO NPs on SRB are discussed separately below.

### 3.2. Zn^2+^ Content Released by ZnO NPs in Solution

ZnO NPs can release a certain concentration of Zn^2+^ in water. Table 3 shows the content of Zn^2+^ in simulated water containing different concentrations of ZnO NPs with or without light treatment. According to Table 1, there was Zn^2+^ dissolved in water containing ZnO NPs, but the concentration was small. When the ZnO NPs’ concentration was lower than 10 mg/L, the Zn^2+^ concentration in water was not more than 0.25 mg/L. Illumination had little effect on the dissolution of Zn^2+^. When the concentration of ZnO NPs was 50 mg/L, the Zn^2+^ concentration in the water with or without light treatment was 0.261 mg/L and 0.263 mg/L, respectively. The release of metal ions is generally believed to be an important factor for the antibacterial properties of metal and metal oxide NPs [27]. Zn^2+^ can penetrate the cell membrane to destroy the protein structure and enter the interior of bacteria to interfere with the activity of enzymes [28]. For general microorganisms, when the Zn^2+^ concentration is in the range of 0.0065~0.65 mg/L, the cells can still be in a normal growth state. However, when the Zn^2+^ concentration exceeds this range, it will affect intracellular homeostasis and protease activity, resulting in cytotoxicity [29]. The data in Table 1 show that in the water containing 0.5~50 mg/L of ZnO NPs, the concentration of Zn^2+^ was no more than 0.263 mg/L, which is a range that does not cause a toxic effect on bacteria [30]. The adaptability of different microorganisms to metal ions is also different. For example, although ZnO NPs have an antibacterial effect on both Gram-positive bacteria and negative bacteria, various research studies demonstrate higher susceptibility and increased sensitivity to Gram-positive bacteria rather than Gram-negative bacteria [31]. This was attributed to several factors, the main factor being the difference in the membrane thickness and membrane ROS sensitivity. Gram-positive bacteria have a membrane and cell wall composed of peptidoglycans, teichoic acid, and lipoteichoic acid that is easier to penetrate compared to the complex cell wall of Gram-negative bacteria, which contains an outer membrane of lipopolysaccharides and a peptidoglycan layer. This prevents the absorption of ROS and ions through the membrane and into the cell [32].

To determine the effect of Zn^2+^ on SRB activity, ZnCl_2_ was added to the bacterial-containing solution without NPs, resulting in a concentration of 0~0.5 mg/L of Zn^2+^. The results are shown in Table 4. When the concentration of Zn^2+^ was no more than 0.25 mg/L, the number of SRB was slightly more than that of the blank, in other words, the presence of Zn^2+^ promoted the growth of microorganisms. This may be because a small amount of Zn^2+^ in the water became a nutrient for SRB [33]. When the concentration of Zn^2+^ was increased to 0.5 mg/L, the bacterial count decreased by only 2.9%. Therefore, it can be considered that within the concentration range of ZnO NPs studied in this paper, the dissolved Zn^2+^ of the NPs may not be the main reason for its antibacterial effect.

### 3.3. ROS Analysis in Water

#### 3.3.1. The Production of ·OH in Water

Terephthalic acid can capture ·OH in water to produce 2-hydroxyterephthalic acid with fluorescence. The fluorescence signal can be captured at the excitation wavelength of 315 nm and the emission wavelength of 425 nm [34]. The level of fluorescence intensity can be used to characterize the amount of ·OH generated [35]. Figure 3 shows the fluorescence spectrum of the simulated water with ZnO NPs. A small amount of ·OH was generated in the water without light treatment, which should be due to the influence of light during sample preparation and sampling [36]. In the water treated with light, the fluorescence intensity corresponding to ·OH was significantly higher than that of the water without light treatment at the same NPs concentration. Under the excitation of light, the electrons in the ZnO NPs transition from the valence band to the conduction band, leaving the valence band empty. The hole robs the hydroxyl electrons in the surrounding environment to form ·OH [37], which can oxidise organic macromolecules, destroy the genetic structure of bacteria, and affect cell development. With the increase in the NPs’ concentration, the fluorescence intensity of the water increased, implying the rise in ·OH content, which improved the antibacterial performance of ZnO NPs on SRB.

#### 3.3.2. The Production of O_2_^·−^ in Water

·OH and O_2_^·−^ are involved in the oxidative stress of microorganisms [38]. Cells can activate antioxidant systems to overcome the destructive effect of ROS on them. However, when excessive ROS is produced, it is difficult to achieve a balance between damage and repair. Cells cannot overcome the effect of excessive ROS, which results in oxidative stress [39]. Table 5 displays the O_2_^·−^ concentration produced in water containing ZnO NPs with and without light treatment. Due to the unavoidable influence of light during water sample preparation and sampling, a small amount of O_2_^·−^ was also detected in the solution without light treatment. Some scholars have also found this phenomenon in their experiments. They thought it might be related to the inability of the sampling environment to be completely dark [36]. Other scholars believed the surface defects of ZnO NPs could also lead to ROS generation even in dark conditions [27]. In the water solution treated with light, the O_2_^·−^ concentration increased with the increase in the NPs’ concentration. The O_2_^·−^ concentration that can be excited by 10 mg/L of ZnO NPs in water was 0.588 mg/L. Under visible light treatment, the amount of ROS generated was significantly increased, which explains the increase in the antibacterial efficiency of NPs under light conditions.

### 3.4. Damage Analysis of Cell Membrane

When nanoparticles come into contact with bacteria, the permeability of the cell membranes increases [40]; the presence of ROS accelerates lipid damage and disrupts cell membrane integrity [16]. When the cell membrane is damaged, DNA and RNA in the cell leak into the solution [41]. These substances have absorption peaks at 260 nm. Therefore, the integrity of the cell membrane can be determined by measuring the absorption [42]. The integrity of the cell membrane is a key factor in determining whether bacteria can grow normally [17]. Figure 4 shows the change in absorbance of bacteria-containing water with and without light treatment at 260 nm. The higher the ZnO NPs concentration, the greater the absorbance of the water sample, indicating that the ZnO NPs increased the rupture of the cell membrane, resulting in more DNA and RNA entering the water. However, the absorbance increased slowly with the increase in the NPs’ concentrations in the system without light treatment. In water containing low concentrations (0.2 mg/L and 0.5 mg/L) of ZnO NPs, the absorbance increased slightly compared with the blank solution, which seemed to contradict the results shown in Figure 2 where the NPs promoted microorganism growth at these concentrations. This may be due to the cell membrane damage caused by ZnO NPs. The released intracellular substances provided the nutrients needed for the growth of other living bacteria. A greater change in the absorbance of the solution was seen after light treatment, indicating that light facilitated the destruction of SRB cell membranes by ZnO NPs.

### 3.5. Effects of ZnO NPs on SRB Biofilm Adhesion on Stainless Steel Surfaces

A layer of biofilm with a certain thickness will be attached to the metal surface after being immersed in the solution containing SRB for a period of time. Figure 5 shows the SEM image of the surface of stainless steel with the attached biofilm in the water containing SRB and different concentrations of ZnO NPs. The biofilm was relatively dense in the water without light treatment. More bacteria adhesion was observed on the stainless-steel surface in water containing 0.5 mg/L of ZnO NPs [Figure 5(b1)] than in the NPs-free water [Figure 5(a1)], and bacterial accumulation and growth are obvious in some areas, indicating that the water containing 0.5 mg/L of ZnO NPs promoted the growth of SRB, resulting in more SRB attached to the stainless steel surface. This is consistent with the results in Figure 2. An explanation for these results is that when the cell membrane of a small part of dead bacteria is damaged, the released substances, such as lipids, proteins, and polysaccharides, can be used as a nutrient source for other living bacteria. At the same time, negligible amounts of ROS are produced in water containing low concentrations of NPs in the absence of light, thus promoting the growth of living bacteria [43]. As the concentration of ZnO NPs increased, a decrease in bacterial attachment was observed, and the ZnO NPs exhibited antibacterial properties. In the water solution without light treatment, the antibacterial effect of ZnO NPs mainly resulted from their contact with SRB due to the low generation of ROS.

Figure 6(a1)–(e1) show that the amount of SRB attached to the stainless-steel surface in light-treated water was relatively low, and the SRB adhesion was scattered. The amount of SRB attached to the stainless-steel surface was consistent with the bacterial content in the water (Figure 2). The ROS produced in water by ZnO NPs are excited by light, which interferes with the normal growth. The antibacterial effect of ZnO NPs in light-treated water lies in the generation of ROS and the contact between NPs and bacteria. Under the action of 10 mg/L of ZnO NPs, only a small amount of bacteria adhered to the stainless-steel surface. Figure 7 shows the EDS results of the red frame area on the surface of Figure 5(c1). It shows that the surface components of stainless steel are mainly C, O, Fe, Cr, Ni, and Zn, which come from the SRB, stainless-steel matrix, and ZnO NPs. Similar results were obtained on stainless-steel surfaces immersed in other water samples containing NPS.

## 4. Conclusions

The effects of the concentration of ZnO nanoparticles and visible light in water on the activity of SRB were researched. In the absence of light, when the concentration of ZnO NPs was not higher than 1 mg/L, the growth of SRB was supported by NPs and a thicker biofilm formed on the stainless-steel surface. With increasing concentrations, the ZnO NPs showed an antibacterial effect on SRB, and the antibacterial efficiency increased with the concentration of NPs. In all water samples treated with light, ZnO NPs showed antibacterial properties, and the antibacterial efficiency was significantly higher than that of the water samples without light treatment.

The antibacterial effect of Zn^2+^ released from ZnO NPs in a low concentration range on SRB was not obvious. In the absence of light, due to the less excited ROS, the antibacterial performance of ZnO NPs mainly came from the contact sterilisation between the NPs and the bacteria. Whereas under light treatment, a certain concentration of ·OH and O_2_^·−^ can be excited by ZnO NPs, which led to further improvement of the antibacterial properties of NPs. The adhesion degree of the SRB biofilm on the stainless-steel surface was consistent with the antibacterial efficiency in water.

## Figures and Tables

**Figure 1 nanomaterials-13-02033-f001:**
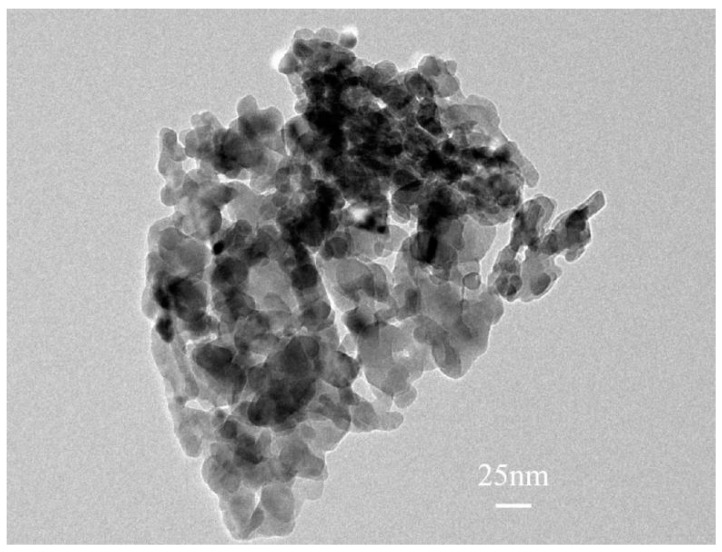
TEM image of ZnO NPs.

**Figure 2 nanomaterials-13-02033-f002:**
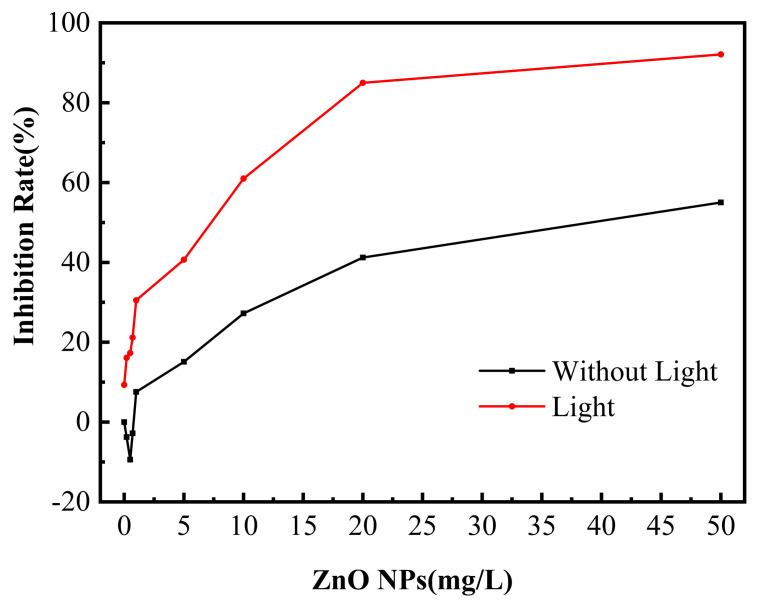
Antibacterial efficiency of ZnO NPs with different concentrations in solution, with or without light treatment.

**Figure 3 nanomaterials-13-02033-f003:**
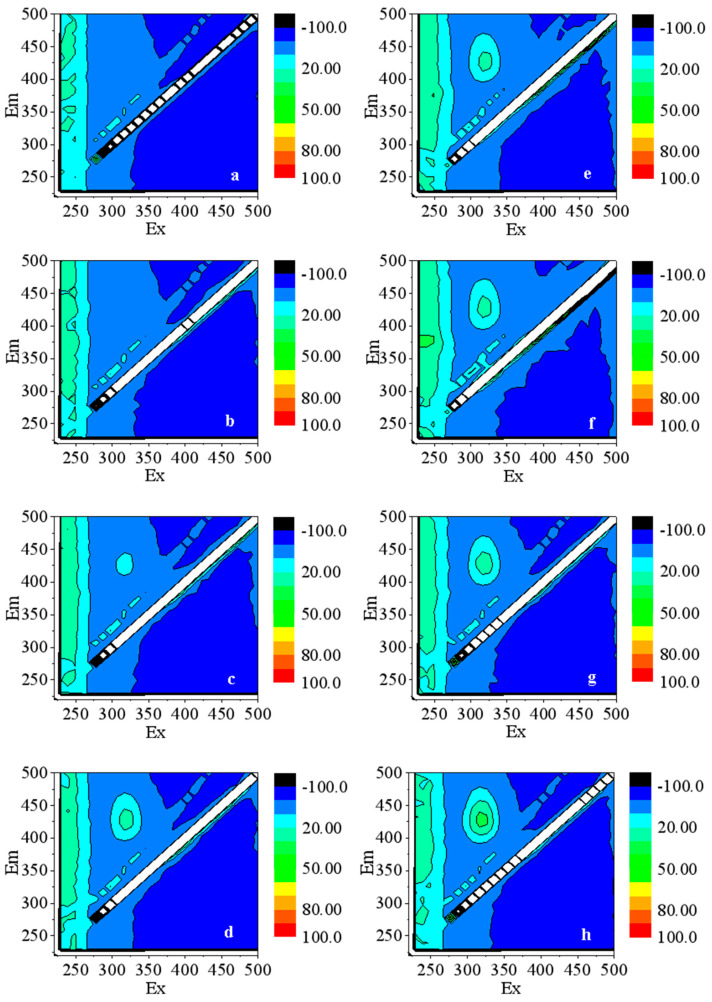
Fluorescence spectra of water samples with ZnO NPs: (**a**–**d**) without light treatment and (**e**–**h**) with light treatment. ZnO NPs concentrations: (**a**,**e**) 0.5 mg/L; (**b**,**f**) 1 mg/L; (**c**,**g**) 5 mg/L; (**d**,**h**) 10 mg/L.

**Figure 4 nanomaterials-13-02033-f004:**
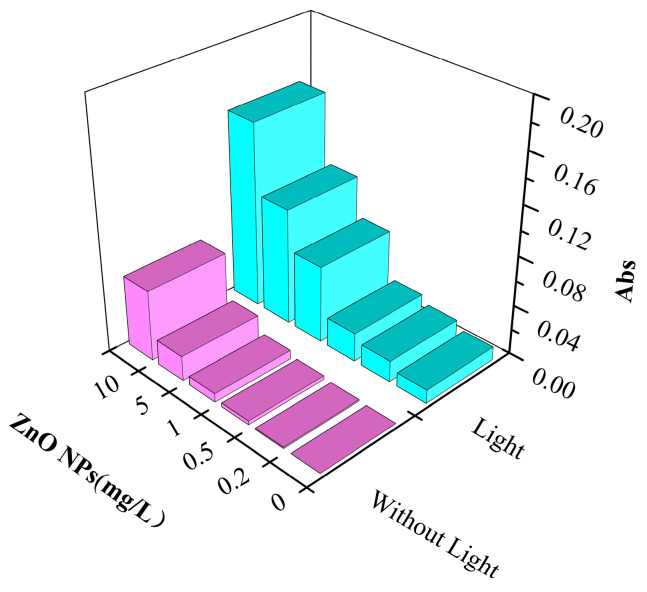
The absorbance of bacteria-containing water with and without light treatment changed with ZnO NPs’ concentration (wavelength 260 nm).

**Figure 5 nanomaterials-13-02033-f005:**
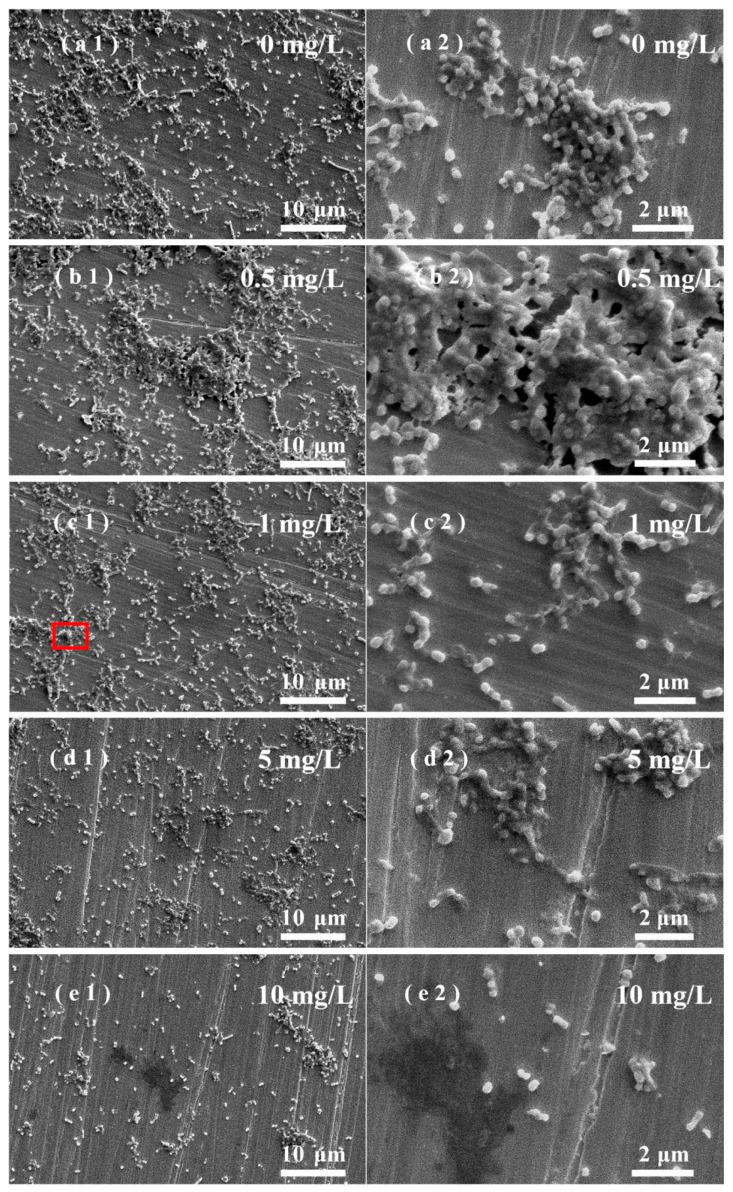
SEM images of biofilm on stainless-steel surfaces in different water solutions without light treatment. ZnO NPs concentration (mg/L): (**a1**,**a2**): 0; (**b1**,**b2**): 0.5; (**c1**,**c2**): 1; (**d1**,**d2**): 5; (**e1**,**e2**): 10.

**Figure 6 nanomaterials-13-02033-f006:**
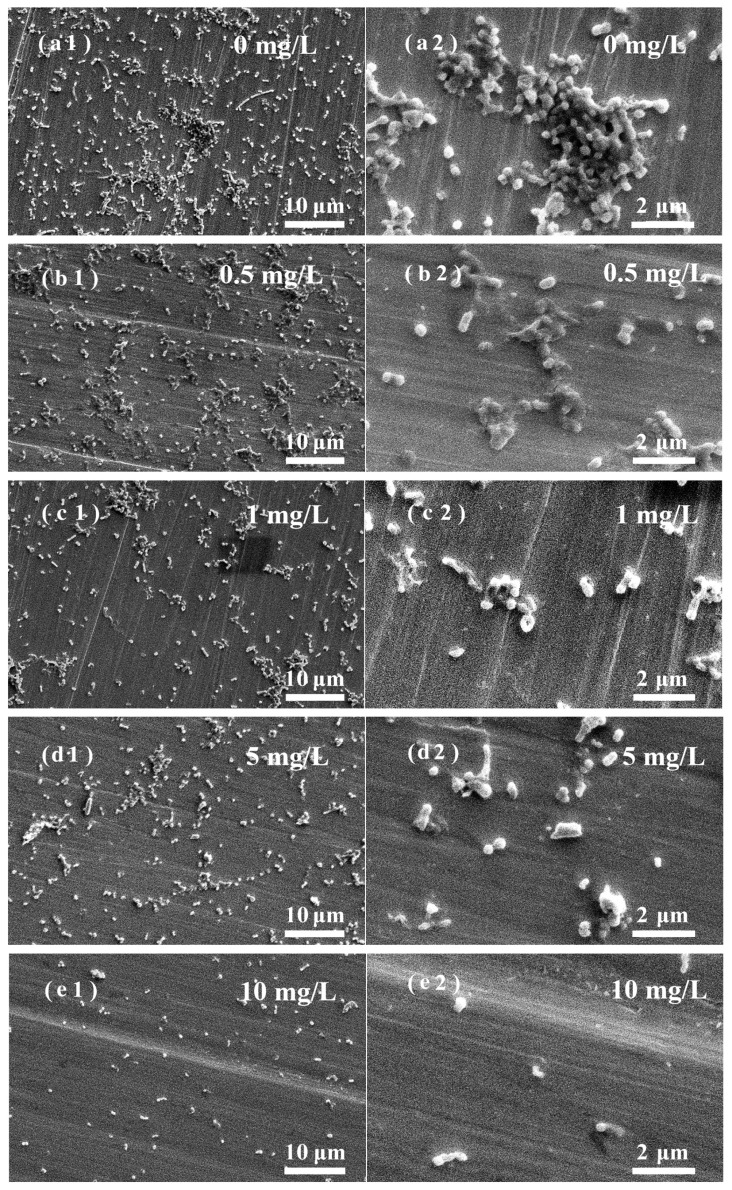
SEM images of biofilm on stainless-steel surfaces in different water solutions with light treatment. ZnO NPs concentration (mg/L): (**a1**,**a2**): 0; (**b1**,**b2**):0.5; (**c1**,**c2**): 1; (**d1**,**d2**): 5; (**e1**,**e2**): 10.

**Figure 7 nanomaterials-13-02033-f007:**
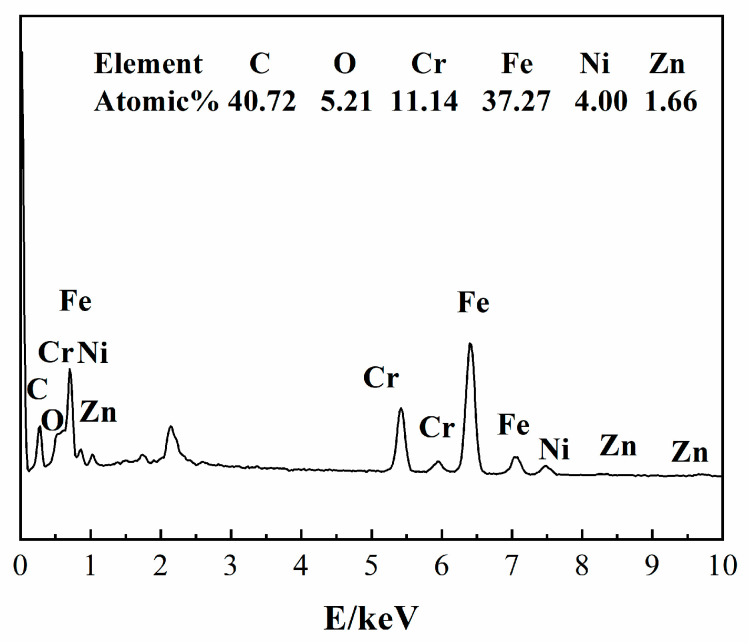
EDS results corresponding to the selected area in Figure 5(c1).

**Table 1 nanomaterials-13-02033-t001:** The composition of the simulated water (mmol/L).

NaCl	NaHCO_3_	Na_2_SO_4_	MgSO_4_	CaCl_2_
7.50	2.00	3.50	0.25	0.50

**Table 2 nanomaterials-13-02033-t002:** Chemical composition of 304 stainless steel.

Element	C	Si	Mn	Cr	Ni	S	P
wt%	0.07	0.59	1.16	17.58	8.25	0.015	0.026

**Table 3 nanomaterials-13-02033-t003:** Concentration of Zn^2+^ released from ZnO NPs in the simulated water.

ZnO NPs (mg/L)		0.5	1	5	10	50
Zn^2+^ concentration (mg/L)	Non-Light treated	0.061	0.146	0.184	0.221	0.261
	Light treated	0.070	0.144	0.180	0.231	0.263

**Table 4 nanomaterials-13-02033-t004:** Effects of different concentrations of Zn^2+^ on SRB activity.

Zn^2+^ (mg/L)	0	0.05	0.25	0.5
Bacteria quantity (CFU/mL)	7.0 × 10^6^	7.3 × 10^6^	7.1 × 10^6^	6.8 × 10^6^

**Table 5 nanomaterials-13-02033-t005:** Concentration of O_2_^·−^ formed in water containing ZnO NPs with and without light treatment.

ZnO (mg/L)		0.5	1	5	10
O_2_^·−^ Concentration (mg/L)	Without Light	0	0.047	0.103	0.137
	With light	0.205	0.239	0.318	0.588

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
