# Peer review of "Antibacterial Effect of Low-Concentration ZnO Nanoparticles on Sulfate-Reducing Bacteria under Visible Light"

_nanomaterials, 2023, doi:10.3390/nano13142033_

Round 1
Reviewer 1 Report
Authors have demonstrated the effect of ZnO NPs with different concentrations in simulated water on the activity of sulfate reducing bacteria (SRB) and their adhesion behavior on stainless steel surfaces with and without visible light treatment. However, there are areas that require revision to meet the publication standards of Nanomaterials. The following comments should be addressed before finalizing the manuscript:
1. The method for calculating the inhibition rate with different concentrations of ZnO NPs in solution, both with and without light treatment as depicted in Figure 2, needs clarification. The authors should provide a detailed explanation of this calculation methodology in the Experimental section.
2. Regarding the statement in section 3.2 regarding the more pronounced antibacterial effect of ZnO NPs on Gram-positive bacteria compared to Gram-negative bacteria, it would be beneficial for the authors to support their explanation with references. For instance, the authors could refer to the reference; Journal of Alloys and Compounds (2021) 876, 160175 to provide additional context and scientific support.
3. In the introduction section, it would be valuable for the authors to discuss the research gap within the field. By highlighting the existing knowledge and identifying the specific gap that their research aims to address, the authors can provide a stronger rationale for their study.
4. The authors should thoroughly review the manuscript for grammar and typographical errors to ensure the clarity and professionalism of the writing.
5. In Figure 6, there should be a space between the numerical value and the unit of measurement, such as "0 mg/L" instead of "0mg/L." Additionally, the authors should consider applying the same standard for spacing in the labels of the SEM images (Figure 5 and 6) to maintain consistency throughout the publication.
Needs moderate modification.
Author Response
See the attachement

Reviewer 2 Report
Reviewer comment
The paper entitled "Antibacterial effect of low concentration ZnO nanoparticles on sulfate-reducing bacteria under visible light" highlighted that the antibacterial performance of ZnO NPs mainly came from the contact sterilization between the NPs and the bacteria due to the less excited ROS on the condition of the absence of light. Whereas under light treatment, a specific concentration of ·OH and O**2 can be exited by ZnO NPs, further improving the antibacterial properties of NPs. The manuscript is well-organized and within the scope of the Nanomaterials. Also, the quality of the manuscript is enough to be published in the Nanomaterials. Therefore, the reviewer agrees to the publication of this manuscript after a few minor revisions.
- The author used commercial ZnO NPs in this research, but there is insufficient information about commercial ZnO NPs. The experimental visible light source (PLS-SXW300 xenon lamp) with a 420 nm filter also showed insufficient information. Therefore, the author should add more information, such as the commercial powder's band gap and the xenon lamp's intensity and power. Especially the band gap of the commercial ZnO NPs plays an essential role in estimating the power of the photocatalysis effect of ZnO NPs. If the author knows the band gap of the ZnO NPs, the author explains the relationship between the production of ROS and the concentration of ZnO NPs more clearly.
- I have little knowledge about the simulated water. The reviewer recommended that the author add more simulated water descriptions to the manuscript.
- Minor check: A couple of paragraphs' font size (Page 5) is bigger than other paragraphs. Please check out the font size of the manuscript.
The overall English grammar of the manuscript is acceptable. The reviewer recommends the minor edition of the English language for double-checking English grammar in the manuscript.
Round 2
Reviewer 1 Report
Authors have not addressed my all queries. Therefore, I would like the authors to go through my following comments again and provide modifications.
Comment 2. Regarding the statement in section 3.2 regarding the more pronounced antibacterial effect of ZnO NPs on Gram-positive bacteria compared to Gram-negative bacteria, it would be beneficial for the authors to support their explanation with references. For instance, the authors could refer to the reference; Journal of Alloys and Compounds (2021) 876, 160175 to provide additional context and scientific support. (Authors have not added this reference in the revised version)
Comment 5. In Figure 6, there should be a space between the numerical value and the unit of measurement, such as "0 mg/L" instead of "0mg/L." Additionally, the authors should consider applying the same standard for spacing in the labels of the SEM images (Figure 5 and 6) to maintain consistency throughout the publication. (please make a gap between number and unit in the standard marking of Fig 5 and 6)
English language should be improved.
